# Longitudinal analyses and predictive factors of radiation-induced lung toxicity-related parameters after stereotactic radiotherapy for lung cancer

**Takaya Yamamoto**[1]\*, **Yoshiyuki Katsuta**[1], **Kiyokazu Sato**[1], **Yoko Tsukita**[2], **Rei Umezawa**[1], **Noriyoshi Takahashi**[1], **Yu Suzuki**[1], **Kazuya Takeda**[1], **Keita Kishida**[1], **So Omata**[1], **Eisaku Miyauchi**[2], **Ryota Saito**[2], **Noriyuki Kadoya**[1], **Keiichi Jingu**[1]

1 Department of Radiation Oncology, Tohoku University Graduate School of Medicine, Sendai, Japan,
2 Department of Respiratory Medicine, Tohoku University Graduate School of Medicine, Sendai, Japan

\* t.yamamoto@rad.med.tohoku.ac.jp

## Abstract

### Background and purpose

The purpose of this prospective study was to investigate changes in longitudinal parameters after stereotactic radiotherapy for lung cancer and to identify possible pretreatment factors related to radiation-induced lung toxicity and the decline in pulmonary function after radiotherapy.

### Materials and methods

Protocol-specified examinations, including 4-D CT, laboratory tests, pulmonary function tests (PFTs) and body composition measurements, were performed before SRT and at 1 month, 4 months and 12 months after stereotactic radiotherapy. Longitudinal differences were tested by using repeated-measures analysis of variance. Correlations were examined by using the Pearson product-moment correlation coefficient (r).

### Results

Sixteen patients were analyzed in this study. During a median follow-up period of 26.6 months, grade 1 and 2 lung toxicity occurred in 11 patients and 1 patient, respectively. The mean Hounsfield units (HU) and standard deviation (SD) of the whole lung, as well as sialy-lated carbohydrate antigen KL-6 (KL-6) and surfactant protein-D (SP-D), peaked at 4 months after radiotherapy (p = 0.11, p<0.01, p = 0.04 and p<0.01, respectively). At 4 months, lung $V_{20\,Gy}$ (%) and $V_{40\,Gy}$ (%) were correlated with changes in SP-D, whereas changes in the mean HU of the lung were related to body mass index and lean body mass index (r = 0.54, p = 0.02; r = 0.57, p = 0.01; r = 0.69, p<0.01; and r = 0.69, p<0.01, respectively). The parameters of PFTs gradually declined over time. When regarding the change in PFTs from pretreatment to 12 months, lung $V_{5\,Gy}$ (cc) showed significant correlations with diffusion capacity for carbon monoxide (DLCO), DLCO/alveolar volume and the relative change in DLCO (r = -0.72, p<0.01; r = -0.73, p<0.01; and r = -0.63, p = 0.01, respectively).

**Funding:** This work was partially supported by the Japan Society for the Promotion of Science KAKENHI [Grande Number 18K15539]. The funder had no role in the study design, data collection and analysis, decision to publish or preparation of the manuscript.

**Competing interests:** TY, YT and RS have received lecturer fees from AstraZeneca KK. EM has received grants from Chugai Pharmaceutical Co Ltd. and Eli Lily Japan KK., honouraria from AstraZeneca KK., Taiho Pharmaceutical Co Ltd., Daiichi Sankyo KK., Boehringer Ingelheim Japan Inc., Bristol Myers Squibb Co Ltd., Novartis Pharma KK., MSD KK., Kyowa Kirin Co Ltd., Merck Biopharma Co Ltd., Pfizer Inc., Ono Pharmaceutical Co Ltd., Otsuka Pharmaceutical Co Ltd. and Towa Pharmaceutical Co. Ltd., and EM has been an advisory board of Chugai Pharmaceutical Co Ltd, Boehringer Ingelheim Japan Inc and Eli Lilly Japan KK. KJ has received consulting fees from Varian Medical Systems and Elekta, and honouraria from Shimazu. Co. YK, KS, RU, NT, YS, KT, KK, SO and NK have no conflicts of interest. This does not alter our adherence to PLOS ONE policies on sharing data and materials.

## Conclusions

The results indicated that some parameters peaked at 4 months, but PFTs were the lowest at 12 months. Significant correlations between lung $V_{5\,Gy}$ (cc) and changes in DLCO and DLCO/alveolar volume were observed.

## Introduction

Systematic therapy for non-small cell lung cancer has shown considerable progress for the past two decades, which is due to the development of new drugs, especially small molecule tyrosine kinase inhibitors and immune checkpoint inhibitors [1–3]. Therefore, systemic therapy is determined by mutations in driver oncogenes and immune checkpoint protein expression, in addition to individual factors. These targeted therapies and immunotherapies are used not only for metastatic lung cancer but also for operable locally advanced lung cancer as neoadjuvant or adjuvant therapies [4, 5]. When regarding early-stage non-small cell lung cancer, surgical resection is a standard treatment [6]. The judgment of operability is performed by thoracic surgeons based on various individual factors, including performance status, frailty, comorbidities and the results of pulmonary function tests (PFTs). In surgical trials, expected postoperative PFTs are used, such as the requirement for an expected postoperative forced expiratory volume in 1 second (FEV1) of 0.8 L or more [7]. Furthermore, stereotactic radiotherapy (SRT), which is often selected as a definitive treatment method for early-stage lung cancer in medically inoperable patients, requires only pretreatment PFTs with sometimes no cutoff value for eligibility being required, which is due to the fact that the post-SRT expectation of PFTs such as surgery is difficult in SRT trials [8]. Additionally, due to the fact that SRT has been performed in such circumstances, the toxicity rates have varied. In the RTOG 0236 trial, 16.3% of the patients had protocol-specified grade 3 or higher toxicity, and an additional 10.9% of the patients had non-protocol-specified high-grade toxicity [9]. The rates of grade 3 or higher adverse events were approximately 10% for patients with noncentral lung cancer (regardless of operability in the subsequent prospective trials); moreover, these rates were 11.9% in the RTOG 0915 trial, 9.6% in pooled analyses for randomized trials for operable patients, 10.6% in the CHISEL trial and 10.3% in the JCOG0403 trial [8–11]. The recent revised STARS trial demonstrated a lower rate of grade 3 or higher adverse events, with only 1 of 80 patients (1.2%) developing grade 3 adverse events [12]. To investigate these differences and the post-SRT effects on PFTs and other factors, more comprehensive analyses are needed. Symptoms such as dyspnea and cough are the complex result of various factors, including a decline in pulmonary function, lung consolidation and systemic or local inflammation. To determine these effects, the primary endpoint of this study was to identify possible pretreatment factors that are related to radiation-induced lung toxicity (RILT) and RILT-related markers at the radiation pneumonitis (RP) phase; in addition, the secondary endpoints were to assess changes in longitudinal parameters after SRT and to identify possible pretreatment factors that have relationships with PFTs at the 12-month follow-up after SRT.

## Materials and methods

### Inclusion criteria and consent from patients

This prospective study was performed to assess RILT and its related factors after SRT for lung cancer. The main inclusion criteria were as follows: pathologically or clinically diagnosed

early-stage primary lung cancer with a tumor diameter of 50 mm or less; agreement between the pulmonologists and radiation oncologists to perform SRT for lung cancer; and no past history of radiotherapy for the thorax. After approval of the Ethical Committee of Tohoku University Hospital (reference number: 2018-2-117), this study was initiated in August 2018 [13]. Written informed consent was obtained from all of the participants. All of the methods were performed in accordance with the Declaration of Helsinki.

## Baseline assessment and follow-up of patients

Protocol-specified examinations, including 4-D CT, laboratory tests, PFTs and body composition measurements, were performed before SRT and at 1 month, 4 months and 12 months after SRT. Follow-up examinations were performed by radiation oncologists and pulmonologists, and adverse events were graded by using the National Cancer Institute Common Terminology Criteria for Adverse Events version 4.0. Four-dimensional CT was performed by using a SOMATOM Definition AS+ CT scanner (Siemens Medical, Iselin, NJ), and 4-DCT images were divided into 10 phases (CT00: end inspiration; CT50: end expiration). Body composition measurements were performed by using the body composition analyzer DC-217A (Tanita, Tokyo, Japan). Lean body mass (LBM) is calculated as the difference between body weight and body fat weight. In addition, body mass index (BMI) is calculated as body weight divided by the square of height in meters, and lean body mass index (LBMI) is calculated as LBM divided by the square of height in meters.

## SRT procedure

Each patient was immobilized in the supine position with a VacQfix Cushion (Qfix, Avondale, PA), after which a planning CT scan at slice intervals of 2 mm was performed. Tumor and organ delineation and radiotherapy planning were performed by using Eclipse (Varian Medical Systems, Palo Alto, CA). Gross tumor volume (GTV) was determined on the basis of the visible extent of the tumor on planning CT, and internal GTV was determined by using 4-D CT images. A planning target volume (PTV) was created by adding 5 mm around the internal GTV for inter- and intrafractional uncertainty. Twenty-eight or 30 Gy in 1 fraction, 48 Gy in 4 fractions or 60 Gy in 8 fractions was prescribed to cover 95% of the PTV by using 6 MV-FFF beams. The planning algorithm included an Acuros XB; in addition, the radiotherapy technique was volumetric modulated arc therapy, and the treatment machine was a TrueBeam STx (Varian Medical Systems, Palo Alto, CA).

## Dosimetric and acquisition data measurements

Lung dose-volume data were used as $V_{n\,Gy}$ (%) or $V_{n\,Gy}$ (cc), which were defined as the percentage or volume of the total lung volume (autosegmented lung minus GTV) exceeding $n$ Gy, respectively. The equivalent dose in 2 Gy fractions (EQD2) was calculated by using a linear-quadratic model with the following formula: $N \times d \times (d + \alpha/\beta)/(2 + \alpha/\beta)$, wherein N is the number of fractions, d is the dose per fraction and $\alpha/\beta$ was applied to 3 Gy for normal lungs based on previous findings [14]. Subsequently, EQD2 was used to analyze $V_{n\,Gy}$ of different SRT schedules, and EQD2 of 5 Gy, 20 Gy and 40 Gy were applied to lung $V_{n\,Gy}$ for dose-volume analyses, due to the fact that we hypothesized that different lung $V_{n\,Gy}$ would affect different parameters that may have different radiosensitivities. Mean Hounsfield units (HU) of the lung were measured via CT50 images by using the autosegmented lung of Eclipse in addition to GTV and consolidation, due to the fact that the distinction between GTV and consolidation after SRT was sometimes difficult to observe. Low attenuation area (LAA) was calculated as the

magnitude of the overlap between the lung and -860 or -960 HU or lower area that was based on previous findings [15].

## Outcomes for assessment and statistical analysis

Time series analyses were performed by using the series of data from each of the assessment points. Any overall longitudinal differences were tested by using repeated-measures analysis of variance. To identify possible predictive factors for RP, changes in the following parameters at the 4-month follow-up after SRT were used: mean HU of the whole lung by using CT50 images, C-reactive protein (CRP), sialylated carbohydrate antigen KL-6 (KL-6) and surfactant protein-D (SP-D). To identify possible predictive factors for the decline in pulmonary function, changes in the following parameters at 12 months after SRT were used: vital capacity (VC, L), forced vital capacity (FVC, L), FEV1 (L), FEV1/FVC (%), FVC1% of predicted, diffusion capacity for carbon monoxide (DLCO, mmol/min/kPa) and DLCO/alveolar volume (VA, L). Delta (Δ) represented the change in each parameter that was calculated as "the observed value minus the baseline value of the parameter", and relative Δ was calculated as "Δ divided by the baseline value". Moreover, DLCO was adjusted for hemoglobin change from baseline. Correlations between predictive pretreatment factors and parameter changes were tested. Additionally, correlations were examined by using the Pearson product-moment correlation coefficient (r). Although multiple testing was performed for some factors (such as lung dose-volume data), a p value less than 0.05 was defined as being statistically significant when considering the exploratory study design. The onset of RILT was calculated from the first day of SRT to the day that an event was confirmed. Cumulative incidences were calculated by using the Kaplan–Meier method, and death was regarded as a competing risk. EZR version 1.54 (Saitama Medical Center, Jichi Medical University, Saitama, Japan), which is a modified version of R commander (R Foundation for Statistical Computing, Vienna, Austria), was used for the analyses [16].

## Results

Seventeen patients were enrolled between August 2018 and February 2021, and one patient later withdrew from the data analysis of the study. As a result, 16 patients were analyzed, and the date of data cutoff was March 31, 2022. Patient characteristics and baseline data are shown in Tables 1 and 2. The pretreatment mean and median (range) values of CRP (mg/dL) were 0.20 and 0.06 (0.01–0.87), respectively. All of the patients completed 12-month examinations at the date of data cutoff, but 1 patient could not undergo spirometry after surgery for oral cancer. Additionally, four sets of 4-D CT images could not be divided into 10 phases. When regarding SRT doses, 3 patients, 2 patients, 6 patients and 5 patients received 28 Gy in 1 fraction, 30 Gy in 1 fraction, 48 Gy in 4 fractions and 60 Gy in 8 fractions, respectively. Details of the PTV and dosimetric parameters are shown in Table 1.

During a median follow-up period of 26.6 months (range: 10.8–37.6 months), one patient died from lung cancer at 16 months after SRT. Thoracic radiotherapy, thoracic surgery or systemic chemotherapy was not performed within 12 months after SRT. Grade 1 or higher RP occurred in 12 patients, including 1 patient with grade 2 RP. Moreover, the one-year cumulative incidences of grade 1 or higher and grade 2 RP were 75.0% (95% confidence interval [CI]: 52.8–92.2%) and 5.2% (95% CI: 0.9–36.8%), respectively. Due to the fact that grade 2 or higher RP occurred in only 1 patient, a predictive factor analysis for RP was not performed.

Longitudinal analyses are shown in Table 2. The mean HU and standard deviation (SD) of the whole lung, KL-6 and SP-D peaked at 4 months after SRT, and these changes were significant, excluding the mean HU of the lung (p = 0.11, p<0.01, p = 0.04 and p<0.01, respectively).

**Table 1. Patient characteristics.**

|  | Distribution or median |
|---|---|
| Age, years | 76 (range: 64–85) |
| Sex | Female: 2, Male: 14 |
| ECOG PS | 0: 12, 1: 4 |
| Charlson comorbidity index | 0–1: 6, 2–3: 9, 4–5:1 |
| Brinkman index | 750 (range: 0–2280) |
| Interstitial lung shadow | Yes: 0, No: 16 |
| Operation history of lung | Yes: 6, No: 10 |
| Pathology | Adenocarcinoma: 6 |
|  | Squamous cell carcinoma: 3 |
|  | Clinically diagnosed: 7 |
| COPD | Yes: 8, No: 8 |
| PTV, cc | 39.2 (range: 13.2–80.3) |
| Lung $V_{5\ Gy}$, % | 14.5 (range: 6.7–31.0) |
| Lung $V_{5\ Gy}$, cc | 491.5 (range: 152.8–823.5) |
| Lung $V_{20\ Gy}$, % | 6.2 (range: 3.1–12.2) |
| Lung $V_{20\ Gy}$, cc | 199.0 (range: 71.2–339.8) |
| Lung $V_{40\ Gy}$, % | 3.2 (range: 1.7–6.0) |
| Lung $V_{40\ Gy}$, cc | 106.1 (range: 45.5–202.9) |

**Table 2. Baseline data and longitudinal analyses of the parameters.**

|  | Before SRT | 1-month FU | 4-month FU | 12-month FU | rANOVA |
|---|---|---|---|---|---|
| Variable | mean ± SD | mean ± SD | mean ± SD | mean ± SD | p value |
| Mean HU value of whole lung | -771 ± 41 | -775 ± 40 | -768 ± 49 | -771 ± 59 | 0.11 |
| SD of HU value of whole lung | 135 ± 10 | 133 ± 8 | 157 ± 35 | 146 ± 19 | <0.01 |
| VC, L | 3.09 ± 0.65 | 3.04 ± 0.65 | 3.01 ± 0.59 | 2.96 ± 0.67 | 0.09 |
| FVC, L | 2.99 ± 0.63 | 2.98 ± 0.65 | 2.87 ± 0.56 | 2.92 ± 0.70 | 0.18 |
| FEV1, L | 1.93 ± 0.48 | 1.89 ± 0.55 | 1.87 ± 0.52 | 1.82 ± 0.53 | 0.02 |
| FEV1/FVC, % | 65.4 ± 12.4 | 63.9 ± 13.6 | 65.9 ± 14.7 | 63.7 ± 17.7 | 0.50 |
| FEV1, % of predicted | 80.6 ± 24.9 | 77.7 ± 27.9 | 78.1 ± 27.1 | 75.7 ± 25.2 | 0.10 |
| DLCO, mmol/min/kPa | 12.6 ± 3.0 | 11.9 ± 4.0 | 11.5 ± 3.1 | 11.4 ± 3.2 | 0.02 |
| DLCO/VA, mmol/min/kPa/L | 3.3 ± 0.8 | 3.2 ± 1.0 | 3.1 ± 0.9 | 3.1 ± 1.0 | 0.21 |
| Neutrophils, mm³ | 3,695 ± 1,348 | 3,421 ± 909 | 3,565 ±1,219 | 3,579 ± 1,639 | 0.87 |
| Lymphocytes, mm³ | 1401 ± 426 | 1,069 ± 350 | 1,204 ± 313 | 1,220 ± 349 | <0.01 |
| Hemoglobin, g/dL | 13.3 ± 1.8 | 13.4 ± 1.7 | 13.0 ± 1.9 | 13.1 ± 2.3 | 0.53 |
| CRP, mg/dL | 0.20 ± 0.24 | 0.51 ± 0.94 | 0.41 ± 0.77 | 0.72 ± 2.04 | 0.39 |
| KL-6, U/mL | 284.3 ± 89.3 | 293.0 ± 114.2 | 332.8 ± 151.9 | 283.6 ± 95.3 | 0.04 |
| SP-D, ng/mL | 104.5 ± 76.5 | 136.0 ± 107.1 | 158.1 ± 109.8 | 112.0 ± 81.3 | <0.01 |
| Body weight, kg | 58.4 ± 13.1 | 57.8 ± 13.1 | 57.3 ± 12.9 | 57.6 ± 14.2 | 0.37 |
| Lean body mass, kg | 46.1 ± 7.6 | 46.2 ± 8.6 | 45.7 ± 7.8 | 45.9 ± 9.0 | 0.86 |

Abbreviations

HU: Hounsfield units, SD: standard deviation, VC: vital capacity, FVC: forced vital capacity, FEV1: forced expiratory volume in 1 second, DLCO: diffusion capacity for carbon monoxide, VA: alveolar volume, CRP: C-reactive protein, KL-6: sialylated carbohydrate antigen KL-6, SP-D: surfactant protein-D, rANOVA: repeated-measures analysis of variance.

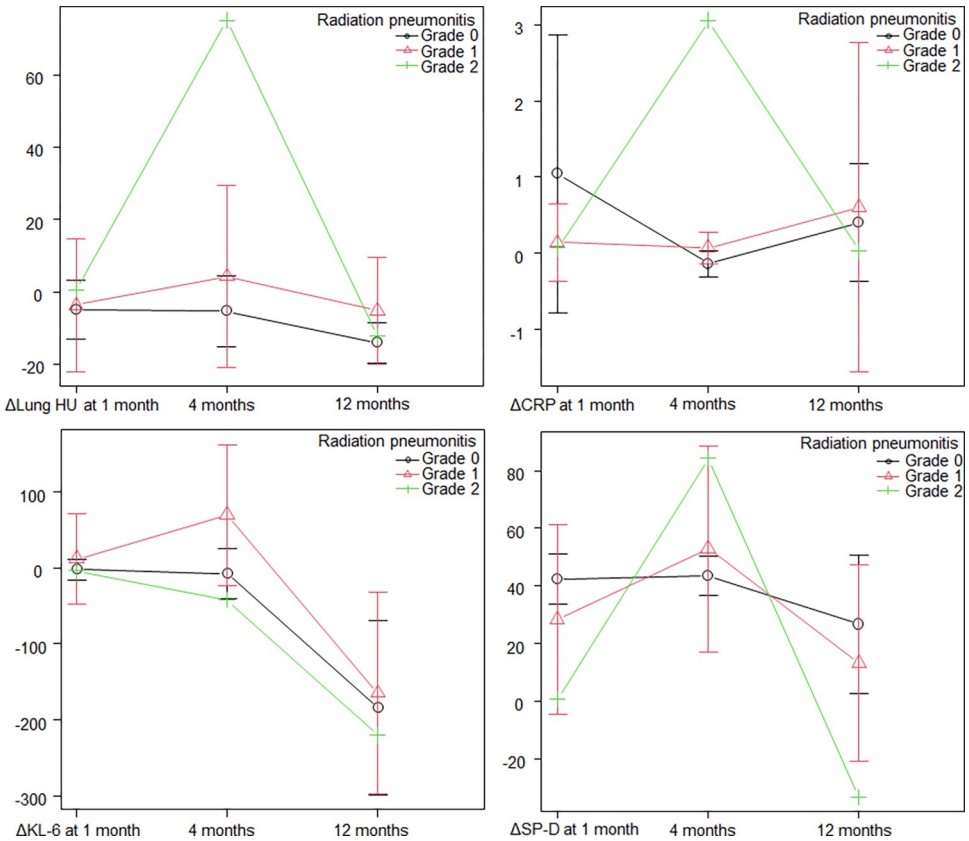

**Fig 1. Changes in the mean HU of the whole lung, CRP, KL-6 and SP-D from baseline to the 1-month, 4-month and 12-month follow-ups according to radiation pneumonitis.** Although Δmean HU of the whole lung, ΔKL-6 and ΔSP-D peaked at 4 months, ΔKL-6 in the patients with grade 2 radiation pneumonitis was decreased at 4 months.

All parameters of PFTs excluding FVC gradually declined over time; therefore, they showed the lowest values at the 12-month follow-up after SRT, but only changes in FEV1 and DLCO were significant (p = 0.02 for both variables). Longitudinal changes in Δmean HU of the whole lung, ΔCRP, ΔKL-6 and ΔSP-D from baseline to each follow-up period according to RP are shown in Fig 1.

Pearson's r for predictive factors of Δmean HU of the whole lung by using CT50 images, ΔCRP, ΔKL-6 and ΔSP-D at the 4-month follow-up after SRT are shown in Table 3. Among these factors, only ΔSP-D had a significant correlation with dosimetric factors: lung $V_{20\ Gy}$ (%) and $V_{40\ Gy}$ (%) (r = 0.54, 95% CI: 0.06-0.81, p = 0.02 and r = 0.57, 95% CI: 0.11-0.83, p = 0.01, respectively). When regarding Δmean HU of the whole lung using CT50 images, ΔCRP and ΔKL-6 showed significant correlations with pretreatment BMI, LBMI or both (Table 3); specifically, BMI was related to Δmean HU of the lung and ΔKL-6 (r = 0.69, 95% CI: 0.26-0.89, p<0.01 and r = 0.54, 95% CI: 0.07-0.82, p = 0.02, respectively), and LBMI was related to Δmean HU of the lung and ΔCRP (r = 0.69, 95% CI: 0.25-0.89, p<0.01 and r = 0.55, 95% CI: 0.07-0.82, p = 0.02, respectively).

Pearson's r of absolute and relative changes in parameters of pulmonary function tests from baseline to 12 months after SRT are shown in Tables 4 and 5. Among dosimetric factors, lung $V_{5\ Gy}$ (cc) showed the strongest correlations with ΔDLCO and ΔDLCO/VA (r = -0.72, 95% CI: -0.90--0.34, p<0.01 and r = -0.73, 95% CI: -0.90--0.35, p<0.01, respectively) and the second strongest correlation with relative ΔDLCO (r = -0.63, 95% CI: -0.86--0.18, p = 0.01). When

**Table 3. Predictive factors for changes in parameters from baseline to 4 months after SRT.**

| Parameters | Δmean HU of lung | | ΔCRP | | ΔKL-6 | | ΔSP-D | |
|---|---|---|---|---|---|---|---|---|
| Predictive pretreatment factors | Pearson's r | p value | Pearson's r | p value | Pearson's r | p value | Pearson's r | p value |
| Lung $V_{5\ Gy}$, % | 0.47 | 0.08 | 0.01 | 0.96 | 0.42 | 0.09 | 0.33 | 0.20 |
| Lung $V_{5\ Gy}$, cc | 0.28 | 0.33 | -0.05 | 0.82 | 0.19 | 0.46 | 0.04 | 0.85 |
| Lung $V_{20\ Gy}$, % | 0.11 | 0.69 | -0.07 | 0.78 | 0.39 | 0.12 | 0.54 | 0.02 |
| Lung $V_{20\ Gy}$, cc | 0.01 | 0.96 | -0.11 | 0.68 | 0.18 | 0.49 | 0.16 | 0.55 |
| Lung $V_{40\ Gy}$, % | -0.05 | 0.85 | -0.05 | 0.85 | 0.26 | 0.32 | 0.57 | 0.01 |
| Lung $V_{40\ Gy}$, cc | -0.13 | 0.65 | -0.10 | 0.70 | 0.07 | 0.79 | 0.15 | 0.57 |
| Age, year-old | 0.18 | 0.52 | -0.10 | 0.68 | 0.13 | 0.61 | 0.29 | 0.26 |
| Brinkman index | 0.32 | 0.25 | -0.03 | 0.89 | 0.05 | 0.85 | -0.09 | 0.72 |
| Charlson comorbidity index | 0.35 | 0.21 | 0.36 | 0.16 | 0.27 | 0.30 | 0.08 | 0.74 |
| Tumor diameter, mm | -0.39 | 0.15 | -0.28 | 0.29 | 0.26 | 0.31 | 0.40 | 0.12 |
| Planning target volume, cc | -0.13 | 0.63 | -0.31 | 0.24 | 0.35 | 0.18 | 0.37 | 0.14 |
| LAA of -860 HU or lower in lung, cc | -0.19 | 0.50 | -0.14 | 0.58 | -0.18 | 0.49 | -0.31 | 0.22 |
| LAA of -960 HU or lower in lung, cc | -0.29 | 0.30 | -0.27 | 0.29 | -0.26 | 0.31 | -0.27 | 0.29 |
| Pretreatment CRP, mg/dL | 0.19 | 0.51 | -0.19 | 0.47 | 0.35 | 0.17 | 0.11 | 0.68 |
| Pretreatment KL-6, U/mL | 0.35 | 0.21 | -0.08 | 0.75 | 0.43 | 0.22 | 0.32 | 0.22 |
| Pretreatment SP-D, ng/mL | 0.17 | 0.54 | -0.06 | 0.81 | 0.19 | 0.48 | 0.44 | 0.08 |
| Pretreatment neutrophil-to-lymphocyte ratio | 0.32 | 0.25 | 0.17 | 0.50 | 0.36 | 0.16 | 0.24 | 0.35 |
| Pretreatment CRP-to-albumin ratio | 0.17 | 0.54 | -0.20 | 0.45 | 0.28 | 0.27 | 0.08 | 0.76 |
| Pretreatment prognostic nutritional index | 0.35 | 0.21 | 0.30 | 0.25 | 0.12 | 0.63 | -0.03 | 0.25 |
| Body weight, kg | 0.25 | 0.37 | 0.15 | 0.55 | 0.29 | 0.26 | 0.13 | 0.60 |
| Body mass index, $kg/m^2$ | 0.69 | <0.01 | 0.15 | 0.57 | 0.54 | 0.02 | 0.35 | 0.17 |
| Lean body mass, kg | 0.32 | 0.25 | 0.04 | 0.87 | 0.07 | 0.76 | 0.04 | 0.87 |
| Lean body mass index, $kg/m^2$ | 0.69 | <0.01 | 0.55 | 0.02 | 0.37 | 0.15 | 0.31 | 0.23 |

Abbreviations

LAA: low attenuation area of the whole lung, Pearson's r: the Pearson product-moment correlation coefficient, other abbreviations are the same as those in Table 2.

regarding ΔVC, ΔFVC, ΔFEV1, ΔFEV1/FVC and ΔFEV1% of predicted, there were no correlations of dosimetric and other pretreatment factors with them, except for correlations between tumor diameter and ΔVC (r = -0.55, 95% CI: -0.83--0.06, p = 0.03) and between LBM and ΔFEV1% of predicted (r = 0.54, 95% CI: 0.04-0.82, p = 0.03). The correlation of ΔFEV1% predicted indicated that patients with a higher pretreatment LBM had a smaller decline in FEV1% predicted at 12 months after SRT. Scatter plots of the highest correlations in Table 4 of ΔVC, ΔFEV1% of predicted, ΔDLCO and ΔDLCO/VA are shown in Fig 2.

## Discussion

In this study, possible RILT-related parameters were prospectively analyzed. Analyses were performed for longitudinal changes in parameters, correlations between possible RILT-related parameters at 4 months after SRT (RP phase) and pretreatment factors, as well as correlations between PFTs at 12 months after SRT and pretreatment factors. RILT consists of these factors and various other factors. Therefore, analyses for each parameter would be beneficial, especially in the current era of radiotherapy combined with immune checkpoint inhibitors to reduce grade 2 or more RILT [17]. In the analyses of longitudinal changes after SRT, the mean HU and SD of the whole lung, KL-6 and SP-D peaked at the 4-month follow-up after SRT, which would reflect the emergence of RP. Prior to the changes in the parameters at the

**Table 4. Predictive factors of absolute changes in parameters of pulmonary function tests from baseline to 12 months after SRT.**

| Parameters | ΔVC | | ΔFVC | | ΔFEV1 | | ΔFEV1/FVC | | ΔFEV1% of predicted | | ΔDLCO | | ΔDLCO/VA | |
|---|---|---|---|---|---|---|---|---|---|---|---|---|---|---|
| **Predictive pretreatment factors** | r | p | r | p | r | p | r | p | r | p | r | p | r | p |
| Lung $V_{5 Gy}$, % | 0.02 | 0.92 | -0.03 | 0.90 | 0.40 | 0.13 | 0.32 | 0.24 | 0.37 | 0.16 | -0.60 | 0.01 | -0.57 | 0.02 |
| Lung $V_{5 Gy}$, cc | -0.02 | 0.94 | 0.11 | 0.67 | 0.24 | 0.38 | 0.06 | 0.82 | 0.31 | 0.25 | -0.72 | <0.01 | -0.73 | <0.01 |
| Lung $V_{20 Gy}$, % | -0.24 | 0.37 | -0.33 | 0.22 | 0.12 | 0.66 | 0.35 | 0.19 | 0.07 | 0.77 | -0.40 | 0.13 | -0.35 | 0.19 |
| Lung $V_{20 Gy}$, cc | -0.30 | 0.26 | -0.21 | 0.44 | -0.01 | 0.95 | 0.14 | 0.61 | 0.01 | 0.95 | -0.65 | <0.01 | -0.63 | 0.01 |
| Lung $V_{40 Gy}$, % | -0.33 | 0.22 | -0.45 | 0.08 | -0.04 | 0.86 | 0.39 | 0.14 | -0.07 | 0.78 | -0.29 | 0.28 | -0.22 | 0.43 |
| Lung $V_{40 Gy}$, cc | -0.36 | 0.17 | -0.33 | 0.22 | -0.08 | 0.76 | 0.25 | 0.36 | -0.05 | 0.83 | -0.50 | 0.05 | -0.49 | 0.05 |
| Age, year-old | 0.15 | 0.59 | 0.15 | 0.58 | -0.09 | 0.72 | -0.21 | 0.44 | 0.12 | 0.66 | 0.03 | 0.89 | -0.10 | 0.71 |
| Brinkman index | 0.07 | 0.79 | 0.24 | 0.37 | 0.06 | 0.81 | -0.10 | 0.70 | 0.36 | 0.18 | -0.23 | 0.39 | -0.02 | 0.93 |
| Charlson comorbidity index | 0.20 | 0.46 | 0.05 | 0.83 | 0.26 | 0.34 | 0.22 | 0.42 | 0.20 | 0.45 | 0.34 | 0.21 | 0.18 | 0.52 |
| Tumor diameter, mm | -0.55 | 0.03 | -0.50 | 0.05 | -0.25 | 0.35 | 0.13 | 0.63 | -0.48 | 0.12 | -0.45 | 0.09 | -0.20 | 0.46 |
| Planning target volume, cc | -0.35 | 0.19 | -0.34 | 0.21 | -0.09 | 0.74 | 0.14 | 0.63 | -0.28 | 0.30 | -0.71 | <0.01 | -0.68 | <0.01 |
| LAA of -860 HU or lower in lung | -0.23 | 0.39 | 0.11 | 0.67 | -0.16 | 0.56 | -0.21 | 0.44 | 0.17 | 0.52 | -0.28 | 0.30 | -0.30 | 0.26 |
| LAA of -960 HU or lower in lung | -0.27 | 0.32 | 0.03 | 0.37 | -0.32 | 0.24 | -0.27 | 0.32 | 0.08 | 0.77 | -0.30 | 0.26 | -0.29 | 0.28 |
| Pretreatment CRP, mg/dL | -0.10 | 0.70 | 0.15 | 0.58 | 0.07 | 0.47 | -0.13 | 0.64 | 0.12 | 0.66 | -0.04 | 0.88 | -0.18 | 0.50 |
| Pretreatment KL-6, U/mL | 0.13 | 0.64 | 0.21 | 0.43 | 0.17 | 0.53 | -0.19 | 0.49 | 0.09 | 0.74 | -0.22 | 0.41 | -0.33 | 0.22 |
| Pretreatment SP-D, ng/mL | -0.25 | 0.35 | -0.28 | 0.30 | -0.37 | 0.16 | -0.02 | 0.92 | -0.40 | 0.13 | 0.07 | 0.78 | 0.04 | 0.88 |
| Pretreatment NLR | 0.24 | 0.37 | 0.19 | 0.48 | 0.47 | 0.07 | 0.05 | 0.84 | 0.31 | 0.25 | -0.34 | 0.21 | -0.52 | 0.04 |
| Pretreatment CAR | -0.09 | 0.73 | 0.17 | 0.54 | 0.04 | 0.88 | -0.15 | 0.58 | 0.11 | 0.67 | -0.06 | 0.80 | -0.20 | 0.46 |
| Pretreatment PNI | 0.23 | 0.40 | 0.28 | 0.30 | 0.09 | 0.72 | -0.16 | 0.56 | 0.04 | 0.87 | 0.39 | 0.14 | 0.39 | 0.14 |
| Body weight, kg | 0.02 | 0.93 | -0.06 | 0.81 | 0.27 | 0.31 | 0.18 | 0.51 | 0.35 | 0.19 | -0.49 | 0.05 | -0.18 | 0.51 |
| Body mass index, $kg/m^2$ | 0.12 | 0.65 | -0.09 | 0.73 | 0.23 | 0.40 | 0.12 | 0.66 | 0.19 | 0.47 | -0.36 | 0.17 | -0.09 | 0.72 |
| Lean body mass, kg | 0.26 | 0.33 | 0.22 | 0.41 | 0.47 | 0.07 | 0.12 | 0.65 | 0.54 | 0.03 | -0.40 | 0.13 | -0.14 | 0.60 |
| Lean body mass index, $kg/m^2$ | 0.48 | 0.06 | 0.27 | 0.31 | 0.47 | 0.07 | 0.02 | 0.93 | 0.39 | 0.14 | -0.21 | 0.43 | -0.02 | 0.91 |

Abbreviations

r: the Pearson product-moment correlation coefficient, p: p value of the Pearson product-moment correlation coefficient, other abbreviations are the same as those in Tables 2 and 3.

4-month follow-up after SRT, a decline in lymphocytes and an increase in SP-D were observed at the 1-month follow-up. Although these early changes are interesting, it is not clear as to whether early changes in the parameters reflect the severity of RILT. Although the mean HU and SD of the whole lung were not changed at the 1-month follow-up in this analysis, an early graphical change after SRT was reported to correlate with severe RP [18]. Another study showed that RP on CT occurred after a median period of 2.9 months after SRT for the symptomatic RP group in comparison with 5.1 months for the asymptomatic RP group [19]. An early emergence of an RP shadow would be needed for a careful follow-up. When regarding CRP, no statistical change was observed, although there was no abnormal distribution of CPR; specifically, the range of pretreatment CRP was 0.01–0.87. It was reported that pretreatment CRP level and maximum CRP after radiotherapy may predict or reflect symptomatic RILT [20, 21]. Actually, the CRP of grade 2 RILT case peaked at 4 months in Fig 1; therefore, the result of this study did not contradict the importance of CRP.

When regarding longitudinal changes in parameters at the 4-month follow-up, only SP-D had relationships with dosimetric parameters, and the relationship between SP-D and lung $V_{40 Gy}$ (%) was the strongest (r = 0.57). Although SP-D peaked at 4 months after SRT in patients with grade 1 and grade 2 RILT, KL-6 in the patient with grade 2 RILT was decreased

**Table 5. Predictive factors of relative parameter changes in pulmonary function tests from baseline to 12 months after SRT.**

| Parameters | Relative ΔVC | | Relative ΔFVC | | RelativeΔFEV1 | | RelativeΔ DLCO | |
|---|---|---|---|---|---|---|---|---|
| **Predictive pretreatment factors** | **r** | **p** | **r** | **p** | **r** | **p** | **r** | **p** |
| Lung V$_{5\ Gy}$, % | -0.04 | 0.88 | -0.05 | 0.85 | 0.25 | 0.36 | -0.40 | 0.13 |
| Lung V$_{5\ Gy}$, cc | -0.02 | 0.93 | 0.10 | 0.70 | 0.09 | 0.72 | -0.63 | 0.01 |
| Lung V$_{20\ Gy}$, % | -0.30 | 0.26 | -0.31 | 0.25 | 0.01 | 0.95 | -0.15 | 0.57 |
| Lung V$_{20\ Gy}$, cc | -0.28 | 0.29 | -0.17 | 0.54 | -0.11 | 0.67 | -0.51 | 0.04 |
| Lung V$_{40\ Gy}$, % | -0.40 | 0.13 | -0.44 | 0.09 | -0.06 | 0.80 | -0.12 | 0.64 |
| Lung V$_{40\ Gy}$, cc | -0.37 | 0.17 | -0.29 | 0.28 | -0.10 | 0.71 | -0.43 | 0.10 |
| Age, year-old | 0.07 | 0.79 | 0.08 | 0.76 | -0.04 | 0.86 | -0.03 | 0.90 |
| Brinkman index | 0.09 | 0.73 | 0.20 | 0.45 | 0.14 | 0.61 | -0.29 | 0.28 |
| Charlson comorbidity index | 0.18 | 0.50 | 0.04 | 0.88 | 0.33 | 0.22 | 0.21 | 0.43 |
| Tumor diameter, mm | -0.50 | 0.05 | -0.41 | 0.12 | -0.25 | 0.35 | -0.40 | 0.13 |
| Planning target volume, cc | -0.36 | 0.17 | -0.30 | 0.27 | -0.18 | 0.51 | -0.65 | <0.01 |
| LAA of -860 HU or lower in lung | -0.13 | 0.64 | 0.11 | 0.68 | -0.22 | 0.42 | -0.50 | 0.05 |
| LAA of -960 HU or lower in lung | -0.19 | 0.49 | 0.02 | 0.93 | -0.40 | 0.13 | -0.51 | 0.04 |
| Pretreatment CRP, mg/dL | -0.08 | 0.76 | 0.11 | 0.68 | 0.03 | 0.89 | -0.22 | 0.42 |
| Pretreatment KL-6, U/mL | 0.11 | 0.67 | 0.21 | 0.43 | 0.08 | 0.76 | -0.31 | 0.25 |
| Pretreatment SP-D, ng/mL | -0.29 | 0.29 | -0.28 | 0.30 | -0.33 | 0.22 | 0.09 | 0.74 |
| Pretreatment NLR | 0.23 | 0.40 | 0.23 | 0.39 | 0.34 | 0.20 | -0.28 | 0.30 |
| Pretreatment CAR | -0.07 | 0.79 | 0.12 | 0.65 | 0.02 | 0.93 | -0.26 | 0.33 |
| Pretreatment PNI | 0.24 | 0.37 | 0.24 | 0.37 | 0.01 | 0.96 | 0.50 | 0.05 |
| Body weight, kg | 0.08 | 0.75 | 0.01 | 0.95 | 0.23 | 0.40 | -0.26 | 0.34 |
| Body mass index, kg/m$^2$ | 0.15 | 0.57 | -0.01 | 0.97 | 0.19 | 0.48 | -0.08 | 0.75 |
| Lean body mass, kg | 0.32 | 0.23 | 0.29 | 0.28 | 0.43 | 0.10 | -0.22 | 0.42 |
| Lean body mass index, kg/m$^2$ | 0.50 | 0.05 | 0.35 | 0.20 | 0.45 | 0.08 | 0.02 | 0.91 |

Abbreviations

Abbreviations are the same as in Tables 2–4.

at the 4-month follow-up, and the results of the studies are therefore different (Fig 1). This discrepancy in the results may be because KL-6 and SP-D have been reported to only exhibit an elevation in 70% of patients with grade 4–5 RILT [22]. At the least, the change in the mean HU of the whole lung peaked at 4 months after SRT. Although the change in mean HU had no correlation with dosimetric factors, this parameter had strong correlations with BMI and LBMI (r = 0.69 and r = 0.69, respectively). The interpretation of these correlations was difficult because BMI was affected by many factors. For example, smoking was associated with lower BMI, but smoking cessation was associated with higher BMI [23]. When patients develop lung emphysema or chronic obstructive pulmonary disease, cachexia can develop [24]. Patients with LAA, which is often caused by chronic obstructive pulmonary disease, tend to show less HU change [15]. There were no significant lung dose-volume correlations with changes in lung HU on CT images, whereas various relationships with quantified CT images have been reported. De Ruysscher et al. reported that they quantified RILT by using the average HU change in each of the irradiated lung dose bins, and they demonstrated that the mean change in HU per Gy was 1.7 ± 2.0 [25]. Individual radiosensitivity differences were also suggested, with HU/Gy ranging from 0 to 10. This difference may also reflect different RILT shadow patterns after radiotherapy [26]. In another study, the normal tissue complication probability of the lung after SRT was investigated by using HU changes in each lung pixel on CT [27]. They reported that D$_{50}$, which was defined as the dose with half of all subjects presenting with a

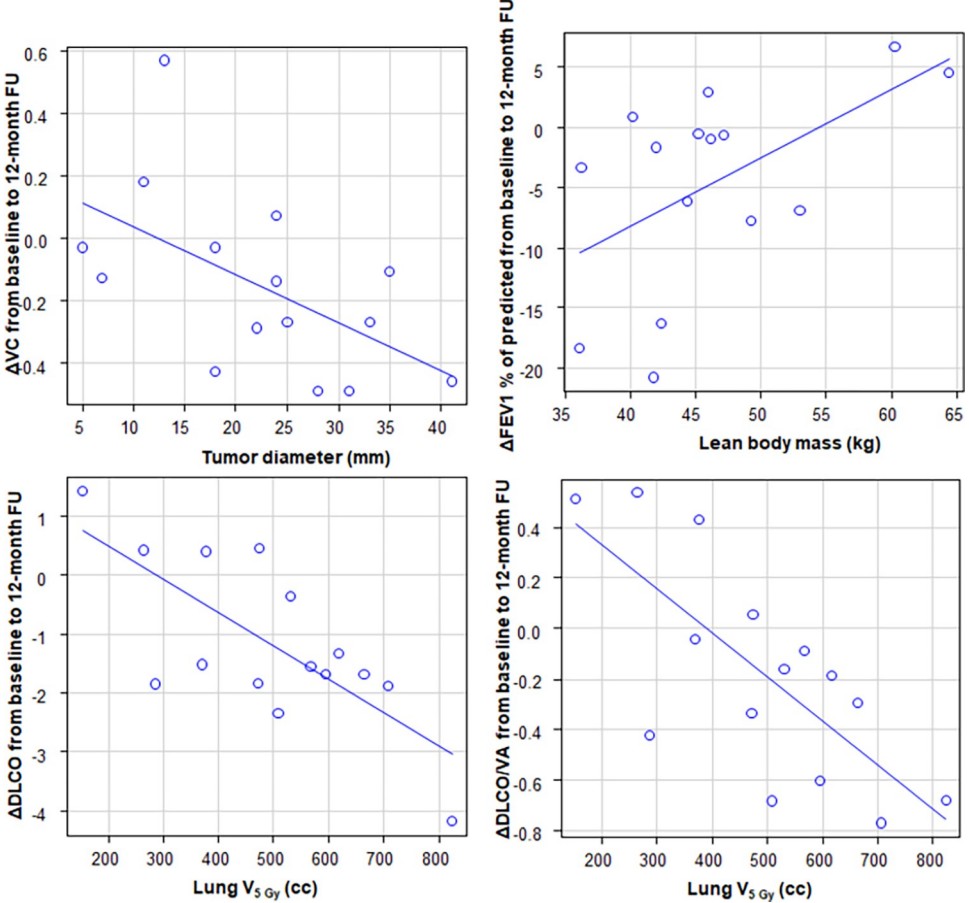

**Fig 2. Scatter plots with regression lines of ΔVC, ΔFEV1% of predicted, ΔDLCO and ΔDLCO/VA with significant correlation parameters.**

symptom, was approximately 35 Gy using 4 NTCP models. Due to the fact that SRT is highly focused radiation, the lung dose-volume effect on the mean HU of the whole lung may be relatively weak, and analyses of local HU changes in the lung will show stronger relationships with lung dose-volume parameters.

When regarding changes in PFT parameters, all of the PFT values gradually declined with time, as was previously reported [28]. Although only FEV1 and DLCO showed significant changes in this study, significant reductions in FVC, FVC % of predicted and FEV1% of predicted after SRT have also been reported [19, 28]. In this study, the median and mean ± SD of relative ΔFEV1 and those of relative ΔDLCO from pretreatment to 12-month follow-up after SRT were -4.2% and -6.8% ± 10.5 and -10.2% and -8.6% ± 11.1, respectively; additionally, those findings were almost the same as those from a previous study [28, 29]. Absolute values are also important because baseline values in SRT patients may be lower than those in surgical candidates, and the median and mean ± SD of ΔFEV1 and those of ΔDLCO were -0.06 L and -0.12 L ± 0.16 and -1.33 and -1.02 ± 1.47, respectively. In surgical patients, it was reported that the mean ± SD values of FEV1 and the transfer factor of the lungs for carbon monoxide (mL/min/mmHg) declined from 2.38 ± 0.79 and 22.3 ± 6.7 to 2.17 ± 0.73 and 21.4 ± 5.9, respectively, at 6 months after lobectomy; additionally, these parameters declined from 2.50 ± 0.47 and 22.8 ± 6.7 to 1.65 ± 0.29 and 16.4 ± 3.7, respectively, at 6 months after pneumonectomy [30]. When considering these results, the magnitude of the reductions in PFT parameters after

SRT was relatively small, but caution is needed when treating patients with extremely low pulmonary function.

Correlations in the parameters of PFTs at the 12-month follow-up are shown in Tables 4 and 5. Lung dose-volume parameters, especially lung $V_{5\ Gy}$ (cc), showed strong correlations with ΔDLCO and ΔDLCO/VA. These results suggested that DLCO may be strongly affected by a low dose distribution in contrast to other PFTs. Therefore, DLCO was thought to have the highest radiosensitivity; additionally, to the best of our knowledge, significant lung dose-volume relationships after SRT with ΔDLCO and ΔDLCO/VA have not been previously reported. Although there were conflicting results for DLCO after SRT, it may be difficult to measure ΔDLCO in a retrospective design because DLCO is a vulnerable parameter and can be affected by chemotherapy agents [31, 32]. Stephans et al. reported correlations of lung ΔFEV1% predicted with lung $V5_{\ Gy}$ (cc) and $V10_{\ Gy}$ (cc), but there was no correlation between ΔDLCO and lung dose-volume parameters in SRT [33]. Some dose-volume relationships with DLCO have been reported in conventional fraction data series. In Hodgkin's lymphoma patients, a lower mean lung dose in patients treated with bleomycin-based chemotherapy alone or bleomycin-based chemotherapy and mediastinal radiotherapy was significantly related to a lower decline in the percentage of predicted DLCO at 1 year [34]. In patients with non-small cell lung cancer treated with 3-dimensional conformal radiotherapy or intensity-modulated radiotherapy or proton beam therapy with or without chemotherapy, several lung dosimetric parameters were significantly correlated with DLCO [35]. In this study, LBM was correlated with ΔFEV1% of predicted significance and ΔFEV1 with marginal significance, but lung dose-volume had no significant correlation. Patients with a high LBM may compensate for the decline in FEV1 and FEV1% predicted by using respiratory muscles, when assuming that patients with a high LBM have increased respiratory muscles. In another study, it was shown that there were significant correlations between lung dose-volume parameters and the relative ΔFEV1 in subgroups of patients divided by the severity of RP [19]. Dose-volume relationships appeared as early as 1 month after SRT in the symptomatic RP subgroup but did not appear until 6 months in the asymptomatic RP subgroup; in addition, there was no significant correlation in the no RP subgroup.

In recent radiotherapy techniques, some lung areas can be intentionally avoided, and there is a new attempt at CT ventilation functional image-guided radiotherapy [36]. A CT ventilation functional image was obtained by 4-D CT images, and this ventilation image had a correlation with FEV1% of predicted and FEV1/FVC [37]. Due to the fact that we can see where it is functional on CT images, functional lung areas can be avoided in radiotherapy planning, and it may be beneficial to avoid functional lung areas leading to a lower incidence of severe RILT. When regarding DLCO, some CT findings have been reported to be associated with DLCO [38]. Although we do not know if there are higher and lower functional areas of DLCO in the lung, further investigations, including radiomics and dosiomics, may aid in visualizing functional DLCO areas in CT images [39, 40]. More precisely, by considering overall pulmonary function, radiotherapy planning can be offered to reduce symptomatic or severe RILT.

There were several limitations in this study. For example, the number of patients and the number of events were relatively small. Only one patient developed grade 2 or more RILT. Due to the fact that this study was an exploratory study, many comparisons were performed; therefore, there may be some false-positives, and further studies are needed to confirm these findings. In addition, the SRT treatment schedule varied; therefore, there was a limitation of dose correction to compare the different radiation schedules. Specifically, there was a limitation of the linear-quadratic model estimation.

In conclusion, different lung dose-volume parameters affected RP-related parameters and parameters of PFTs. Some possible RILT markers peaked at 4 months, but PFTs declined over

time and were the lowest at 12 months. The analyses of PFT parameter changes from pretreatment to 12 months after SRT found that DLCO and DLCO/VA were significantly correlated with lung $V_{5 Gy}$ (cc), which indicated that DLCO and DLCO/VA were possibly affected by a relatively lower radiation dose distribution.

## Supporting information

**S1 File.**
(XLSX)

## Acknowledgments

We are grateful to the radiation oncologists, pulmonologists, medical physicists and radiation technologists at Tohoku University Hospital who contributed to the recruitment, treatment and follow-up of the patients. When creating the protocol, advice was provided to us by Dr. Soichiro Toda from the Clinical Research, Innovation and Education Center, Tohoku University Hospital.

## Author Contributions

**Conceptualization:** Takaya Yamamoto.

**Data curation:** Takaya Yamamoto, Yoshiyuki Katsuta.

**Formal analysis:** Takaya Yamamoto.

**Funding acquisition:** Takaya Yamamoto.

**Investigation:** Takaya Yamamoto, Yoshiyuki Katsuta, Kiyokazu Sato, Yoko Tsukita, Rei Umezawa, Noriyoshi Takahashi, Yu Suzuki, Kazuya Takeda, Keita Kishida, So Omata, Eisaku Miyauchi, Ryota Saito, Noriyuki Kadoya, Keiichi Jingu.

**Methodology:** Takaya Yamamoto.

**Project administration:** Takaya Yamamoto, Keiichi Jingu.

**Resources:** Takaya Yamamoto, Kiyokazu Sato, Yoko Tsukita, Eisaku Miyauchi, Ryota Saito, Noriyuki Kadoya.

**Supervision:** Yoshiyuki Katsuta, Keiichi Jingu.

**Validation:** Takaya Yamamoto, Noriyuki Kadoya, Keiichi Jingu.

**Writing – original draft:** Takaya Yamamoto.

**Writing – review & editing:** Takaya Yamamoto, Yoshiyuki Katsuta, Kiyokazu Sato, Yoko Tsukita, Rei Umezawa, Noriyoshi Takahashi, Yu Suzuki, Kazuya Takeda, Keita Kishida, So Omata, Eisaku Miyauchi, Ryota Saito, Noriyuki Kadoya, Keiichi Jingu.

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
