## [Decision Letter · Decision Letter 0]

9 Nov 2022

PONE-D-22-26386Longitudinal analyses and predictive factors of radiation-induced lung toxicity-related parameters after stereotactic radiotherapy for lung cancerPLOS ONE

Dear Dr. Yamamoto,

Thank you for submitting your manuscript to PLOS ONE. After careful consideration, we feel that it has merit but does not fully meet PLOS ONE’s publication criteria as it currently stands. Therefore, we invite you to submit a revised version of the manuscript that addresses the points raised during the review process.

We look forward to receiving your revised manuscript.

Kind regards,

Alessandro Rizzo

Academic Editor

PLOS ONE

Journal Requirements:

"This work was supported in part by Japan Society for the Promotion of Science KAKENHI [Grande Number 18K15539]."

"TY, YT and RS have received lecturer fees from AstraZeneca KK.

EM has received grants from Chugai Pharmaceutical Co Ltd. and Eli Lily Japan KK., honouraria from AstraZeneca KK., Taiho Pharmaceutical Co Ltd., Daiichi Sankyo KK., Boehringer Ingelheim Japan Inc., Bristol Myers Squibb Co Ltd., Novartis Pharma KK., MSD KK., Kyowa Kirin Co Ltd., Merck Biopharma Co Ltd., Pfizer Inc., Ono Pharmaceutical Co Ltd., Otsuka Pharmaceutical Co Ltd. and Towa Pharmaceutical Co. Ltd., and EM has been an advisory board of Chugai Pharmaceutical Co Ltd, Boehringer Ingelheim Japan Inc and Eli Lilly Japan KK. 

KJ has received consulting fees from Varian Medical Systems and Elekta, and honouraria from Shimazu. Co. 

YK, KS, RU, NT, YS, KT, KK, SO and NK have no conflicts of interest."

Reviewers' comments:

Reviewer's Responses to Questions

**Comments to the Author**

1. Is the manuscript technically sound, and do the data support the conclusions?

Reviewer #1: Partly

Reviewer #2: Yes

2. Has the statistical analysis been performed appropriately and rigorously? 

Reviewer #1: Yes

Reviewer #2: No

3. Have the authors made all data underlying the findings in their manuscript fully available?

Reviewer #1: Yes

Reviewer #2: Yes

4. Is the manuscript presented in an intelligible fashion and written in standard English?

Reviewer #1: No

Reviewer #2: No

5. Review Comments to the Author

Reviewer #1: Dear Editor, thank you so much for inviting me to revise this manuscript about lung cancer.

This study addresses a current topic. However, important limitations should be acknowledged.

The manuscript is quite well written and organized. English should be improved.

Figures and tables are comprehensive and clear.

The introduction explains in a clear and coherent manner the background of this study.

We suggest the following modifications:

• Introduction section: although the authors correctly included important papers in this setting, we believe the systemic treatment scenario for lung cancer should be further described and some recently published studies should be cited within the introduction ( PMID: 35326555; PMID: 33053439 ; PMID: 35029065), only for a matter of consistency. We think it might be useful to introduce the topic of this interesting study.

• Methods and Statistical Analysis: nothing to add.

• Discussion section: Very interesting and timely discussion. Of note, the authors should expand the Discussion section, including a more personal perspective to reflect on. For example, they could answer the following questions – in order to facilitate the understanding of this complex topic to readers: what potential does this study hold? What are the knowledge gaps and how do researchers tackle them? How do you see this area unfolding in the next 5 years? We think it would be extremely interesting for the readers.

However, we think the authors should be acknowledged for their work. In fact, they correctly addressed an important topic, the methods sound good and their discussion is well balanced.

One additional little flaw: the authors could better explain the limitations of their work, in the last part of the Discussion. Among these, the sample size is very limited, something that precludes from making strong statements.

We believe this article is suitable for publication in the journal although major revisions are needed. The main strengths of this paper are that it addresses an interesting and very timely question and provides a clear answer, with some limitations.

We suggest a linguistic revision and the addition of some references for a matter of consistency. Moreover, the authors should better clarify some points.

Reviewer #2: Manuscript number: PONE-D-22-26386

Research Article entitled “Longitudinal analyses and predictive factors of radiation-induced lung toxicity-related parameters after stereotactic radiotherapy for lung cancer” by Takaya Yamamoto and his group, reports the changes in longitudinal parameters after stereotactic radiotherapy for lung cancer and demonstrates the possible pretreatment risk factors that are related to radiation-induced lung toxicity and pulmonary function decrease. The approach is quite simple. The manuscript is understandably and systematically written, the reader can follow the results and discussion.

Nevertheless, I have some remarks, which in my opinion, authors should consider and improve the manuscript accordingly. The manuscript should be edited for the English language. Grammatical errors have to be checked (verbs, prepositions, articles). Some sentences have to be rewritten or added. In my opinion, there is a main concern:

In the case of the values of CRP (mg/dL) in Table 2, why are the standard deviations (SD) of all cases higher than the mean values, respectively? Actually, this might happen when there is high variation between values and an abnormal distribution of data is present for the analysis. Similarly, the SDs for Figure 1 are so large that they have covered the indicated main values of all four graphs. Perhaps it should not be considered an appropriate way to show the variability because the data seems not well distributed.

Recommendation: While the overall approach and scientific conclusions are generally sound. I would like to suggest that this article will be published in PLOS ONE after minor revision, particularly in the grammar and style, are needed to prepare the manuscript for publication.

Here are some comments as follows.

1. The abstract should be improved, especially the part of the conclusion. For clarity, I also suggest fewer abbreviations in the abstract. I miss the statement of the novelty of the present study.

2. Page: 2, Line: 27; Article is missing before “decline”.

3. Page: 5, Line: 108; Use same format for all. For example, you should write "28" instead of "Twenty-eight".

4. Page: 6, Line: 130, 132; Articles are missing before “series”, and “following”.

5. Page: 10, Line: 179; “are shown” instead of ‘showed’.

6. Page: 10, Line: 180; significant “,” excluding “the” mean….

7. Page: 17, Line: 232; Article is missing before “early”.

8. Page: 18, Line: 248; Article is missing before “change”.

9. Page: 18, Line: 250; “these correlations” instead of ‘this correlations’.

10. Page: 18, Line: 265; “for the decline in” instead of ‘for decline of’.

11. Page: 20, Line: 313; “further studies were” instead of ‘further study were’.

12. Page: 21, Line: 318; “DLCO and DLCO/VA were affected” instead of ‘DLCO and DLCO/VA was affected.

Also, the discussion needs to be rephrased in order to comment on the results with other research rather than quoting the results. Missing statement about the novelty and the importance of research.

6. PLOS authors have the option to publish the peer review history of their article (what does this mean?). If published, this will include your full peer review and any attached files.

Reviewer #1: No

Reviewer #2: No

---

## [Author Response · Author response to Decision Letter 0]

19 Nov 2022

Reply for reviewers

For Reviewer #1

The manuscript is quite well written and organized. English should be improved.

Figures and tables are comprehensive and clear.

The introduction explains in a clear and coherent manner the background of this study.

Response:

Thank you for your comments. We improved the English using professional proofreading. We enclosed certification of English professional proofreading.

• Introduction section: although the authors correctly included important papers in this setting, we believe the systemic treatment scenario for lung cancer should be further described and some recently published studies should be cited within the introduction ( PMID: 35326555; PMID: 33053439 ; PMID: 35029065), only for a matter of consistency. We think it might be useful to introduce the topic of this interesting study.

Response: Thank you very much for your advice. We added the systemic treatment scenario in introduction section and addressed these manuscripts.

“Systematic therapy for non-small cell lung cancer has shown considerable progress for the past two decades, which is due to the development of new drugs, especially small molecule tyrosine kinase inhibitors and immune checkpoint inhibitors [1-3]. Therefore, systemic therapy is determined by mutations in driver oncogenes and immune checkpoint protein expression, in addition to individual factors. These targeted therapies and immunotherapies are used not only for metastatic lung cancer but also for operable locally advanced lung cancer as neoadjuvant or adjuvant therapies [4,5]. When regarding early-stage non-small cell lung cancer, surgical resection is a standard treatment [6]”

• Discussion section: Very interesting and timely discussion. Of note, the authors should expand the Discussion section, including a more personal perspective to reflect on. For example, they could answer the following questions – in order to facilitate the understanding of this complex topic to readers: what potential does this study hold? What are the knowledge gaps and how do researchers tackle them? How do you see this area unfolding in the next 5 years? We think it would be extremely interesting for the readers.

However, we think the authors should be acknowledged for their work. In fact, they correctly addressed an important topic, the methods sound good and their discussion is well balanced.

Response: Thank you for your comments. Although it was tough theme, we discussed and added following paragraph in discussion section.

“In recent radiotherapy techniques, some lung areas can be intentionally avoided, and there is a new attempt at CT ventilation functional image-guided radiotherapy [36]. A CT ventilation functional image was obtained by 4-D CT images, and this ventilation image had a correlation with FEV1% of predicted and FEV1/FVC [37]. Due to the fact that we can see where it is functional on CT images, functional lung areas can be avoided in radiotherapy planning, and it may be beneficial to avoid functional lung areas leading to a lower incidence of severe RILT. When regarding DLCO, some CT findings have been reported to be associated with DLCO [38]. Although we do not know if there are higher and lower functional areas of DLCO in the lung, further investigations, including radiomics and dosiomics, may aid in visualizing functional DLCO areas in CT images [39,40]. More precisely, by considering overall pulmonary function, radiotherapy planning can be offered to reduce symptomatic or severe RILT.”

• One additional little flaw: the authors could better explain the limitations of their work, in the last part of the Discussion. Among these, the sample size is very limited, something that precludes from making strong statements.

Response: Thank you for your comments. We modified some phrases of strong statements.

We believe this article is suitable for publication in the journal although major revisions are needed. The main strengths of this paper are that it addresses an interesting and very timely question and provides a clear answer, with some limitations.

Response: Thank you very much for your heartful comments.

 

For Reviewer #2

Thank you very much for your comments. Here is our point-by-point responses to your comments and concerns.

1. The manuscript should be edited for the English language. Grammatical errors have to be checked (verbs, prepositions, articles). Some sentences have to be rewritten or added.

Response:

Thank you for your advice. Because we are not English native speakers, we revised the English using professional proofreading. We enclosed certification of English professional proofreading.

2. In my opinion, there is a main concern: In the case of the values of CRP (mg/dL) in Table 2, why are the standard deviations (SD) of all cases higher than the mean values, respectively? Actually, this might happen when there is high variation between values and an abnormal distribution of data is present for the analysis. Similarly, the SDs for Figure 1 are so large that they have covered the indicated main values of all four graphs. Perhaps it should not be considered an appropriate way to show the variability because the data seems not well distributed.

Response:

Thank you for your question. Yes, the SD of CRP was higher than the mean value of CRP and this is thought to be the feature of CRP. Firstly, such distribution of CRP was reported in much more patients. Yao et al. reported that mean ± SD of CRP (mg/L) was 14.92 ± 23.45 in 182 advanced lung cancer patients. (Yao Y, et al. Cancer Immunol Immunother. 2013;6:471-9.). In the healthy non-smokers and current smokers report, mean ± SD of CRP (mg/dL) were 0.1 ±0.4 and 0.1 ± 0.3, respectively (Ohshimo S, et al. Sarcoidosis Vasc Diffuse Lung Dis. 2009;26:47-53.). In the stereotactic reports, Tsurugai et al. reported that “median (range)” of CRP in 42 patients with idiopathic interstitial pneumonias and 466 patients without idiopathic interstitial pneumonias were 0.2 (0–3.7) and 0.1 (0–8.9), respectively. (Tsurugai Y, et al. Radiother Oncol. 2017;12:310-316.). We think that the distribution of CRP in this study is natural. Secondly, although there is the concern about an abnormal distribution of data, the rage of pretreatment CRP (mg/dL) was 0.01-0.87. CRP <10 mg/L was one of the requirements for good prognosis of modified Glasgow prognostic score (prognostic score of lung cancer and other cancers; Jin J, et al. PLoS One. 2017;12:e0184412.). Because CRP 10 mg/L was equal to CRP 1.0 mg/dL, all pretreatment CRP of this study fulfilled this cut-off value. Therefore, we think there is no abnormal distribution of data in this study. Finally, figure 1 showed larger SD of CRP, this was because CRP showed dynamic change after treatment. The following figure was reported by Liu et al (Liu F, et al. Int J Radiat Oncol Biol Phys. 2022;114:433-443.). They reported longitudinal CRP data after chemoradiotherapy for advanced lung cancer. Although CRP showed mg/L scale, quantile (boxes) and rage (error bars) of CRP was very large. Therefore, we think that to show large SD in figure 1 is also important because this is thought to be one of the feature of CRP.

We think your concern is quite natural, we added more information about CRP in the manuscript to reduce these concerns.

3. While the overall approach and scientific conclusions are generally sound. I would like to suggest that this article will be published in PLOS ONE after minor revision, particularly in the grammar and style, are needed to prepare the manuscript for publication.

1. The abstract should be improved, especially the part of the conclusion. For clarity, I also suggest fewer abbreviations in the abstract. I miss the statement of the novelty of the present study.

Also, the discussion needs to be rephrased in order to comment on the results with other research rather than quoting the results. Missing statement about the novelty and the importance of research.

Response:

Thank you for your advices. We agreed what you mentioned, and we modified the abstract section and discussion section.

3. Page: 5, Line: 108; Use same format for all. For example, you should write "28" instead of "Twenty-eight".

Response:

Thank you for your comments. In manuscript, to spell out the number would be common at the beginning of a sentence, therefore, we did not modify this point.

2. Page: 2, Line: 27; Article is missing before “decline”.

4. Page: 6, Line: 130, 132; Articles are missing before “series”, and “following”.

5. Page: 10, Line: 179; “are shown” instead of ‘showed’.

6. Page: 10, Line: 180; significant “,” excluding “the” mean….

7. Page: 17, Line: 232; Article is missing before “early”.

8. Page: 18, Line: 248; Article is missing before “change”

9. Page: 18, Line: 250; “these correlations” instead of ‘this correlations’.

10. Page: 18, Line: 265; “for the decline in” instead of ‘for decline of’.

11. Page: 20, Line: 313; “further studies were” instead of ‘further study were’.

12. Page: 21, Line: 318; “DLCO and DLCO/VA were affected” instead of ‘DLCO and DLCO/VA was affected.

Response:

Thank you very much for your advices. We modified sentences.

---

## [Editor Report · Decision Letter 1]

22 Nov 2022

Longitudinal analyses and predictive factors of radiation-induced lung toxicity-related parameters after stereotactic radiotherapy for lung cancer

PONE-D-22-26386R1

Dear Dr. Yamamoto,

We’re pleased to inform you that your manuscript has been judged scientifically suitable for publication and will be formally accepted for publication once it meets all outstanding technical requirements.

Kind regards,

Alessandro Rizzo

Academic Editor

PLOS ONE

---

## [Editor Report · Acceptance letter]

24 Nov 2022

PONE-D-22-26386R1 

Longitudinal analyses and predictive factors of radiation-induced lung toxicity-related parameters after stereotactic radiotherapy for lung cancer 

Dear Dr. Yamamoto:

I'm pleased to inform you that your manuscript has been deemed suitable for publication in PLOS ONE. Congratulations! Your manuscript is now with our production department. 

Kind regards, 

on behalf of

Dr. Alessandro Rizzo 

Academic Editor

PLOS ONE